

# Enhanced biomass and thermotolerance of *Arabidopsis* by *SiERECTA* isolated from *Setaria italica* L

Jiacheng Zheng[1,2,*], Xiaoyi Huang[1,*], Jieqin Li[1], Qingyuan He[1], Wan Zhao[2], Chaowu Zeng[3], Haizhou Chen[4], Qiuwen Zhan[1] and Zhaoshi Xu[2]

[1] Anhui Science and Technology University, College of Agronomy, Fengyang, Anhui, China
[2] Chinese Academy of Agricultural Sciences (CAAS)/National Key Facility for Crop Gene Resources and Genetic Improvement, Key Laboratory of Biology and Genetic Improvement of Triticeae Crops, Ministry of Agriculture, Institute of Crop Science, Beijing, China
[3] Xinjiang Academy of Agricultural Sciences, Institute of Crop Sciences, Urumuqi, Xinjiang, China
[4] Anhui Youxin Agricultural Science and Technology Co. LTD, Hefei, Anhui, China
[*] These authors contributed equally to this work.

Corresponding authors
Qiuwen Zhan, qwzhan@163.com, 18755065573@126.com
Zhaoshi Xu, xuzhaoshi@caas.cn

## ABSTRACT

Foxtail millet is commonly used as a food and forage grass. ERECTA (ER) is a receptor-like kinase that can improve plant biomass and stress resistance. The sorghum *SbER10_X1* gene was used as a probe to identify *ER* family genes on the *Setaria italica* genomes (*SiERs*), and determine the characteristics of the SiERs family. Herein, the structural features, expression patterns, and thermotolerance of SiERs function were identified by bioinformatics analysis, real-time PCR and transgenesis estimation. Results showed that *SiERs* had four members: two members were located on chromosome 1 with a total of six copies (*SiER1_X1*, *SiER1_X2*, *SiER1_X3*, *SiER1_X4*, *SiER1_X5*, and *SiER1_X6*), and two were on chromosome 4, namely, *SiER4* (*SiER4_X1* and *SiER4_X2*) and *SiERL1*. Among them, *SiER1_X4* and *SiER4_X1* were expressed highest in above-ground organs of foxtail millet, and actively responded to treatments with abscisic acid, brassinolide, gibberellin, and indole acetic acid. After overexpression of *SiER1_X4* and *SiER4_X1* in *Arabidopsis*, the plant height and biomass of the transgenic *Arabidopsis* significantly increased. Following high-temperature treatment, transgenic seedlings survived better compared to wild type. Transgenic lines showed higher SOD and POD activities, and expression level of *AtHSF1* and *AtBl1* genes significantly increased. These results indicated that SiER1_X4 and SiER4_X1 played important regulatory roles in plant growth and thermotolerance. The two genes provide potential targets for conventional breeding or biotechnological intervention to improve the biomass of forage grass and thermotolerance of field crops.

## INTRODUCTION

Foxtail millet is an annual C$_4$ crop that can be used as a food and forage grass (*Singh, Muthamilarasan & Prasad, 2021*). In arid and semi-arid regions, foxtail millet shows strong tolerance to various abiotic stresses such as drought, salinity, high temperature (*Aidoo*

*et al., 2016*). However, the natural conditions experienced by field crops are increasingly complex. In addition, the human demand for food and energy is intensified by the global population growth and per capita income increase. To cope with this severe challenge, crop varieties need to be improved by traditional breeding, functional gene screening, genome editing, or other technologies. The foxtail millet genome contains many excellent genes related to drought resistance, high yield, and high light efficiency (*Zhang et al., 2012*). The rational utilization of foxtail millet functional genes is an important strategy to ensure future food security.

ERECTA (ER) belongs to the receptor-like kinases (RLKs), which are involved in the regulation of plant photosynthesis and transpiration efficiency, thereby increasing biomass and plant resistance (*Masle, Gilmore & Farquhar, 2005*; *Van Zanten et al., 2009*). Overexpression of *SbER2-1* in maize conferred increased drought tolerance, especially in regard to improved water-use efficiency (*Li et al., 2019*). When the *Arabidopsis AtER* gene was overexpressed in tomato and rice, the biomass of transgenic lines was increased and heat tolerance was enhanced (*Shen et al., 2015*). Further studies have shown that the fusion gene of chitin elicitor receptor kinase 1 and *ER* (*CERK1n-ER*) can induce the production of chitooligosaccharides and improve heat tolerance of *Arabidopsis* (*Chen et al., 2020*). Overexpression of poplar *PdER* gene in *Arabidopsis* resulted in reduced stomatal density, thereby influencing transpiration, water-use efficiency and drought tolerance (*Li et al., 2021*). Interference of MAPK cascade reaction through the interaction of *ER* with the *BAK1* gene increased the resistance of *Arabidopsis* to the necrotrophic fungus, *Plectosphaerella cucumerina BMM* (*PcBMM*) (*Jorda et al., 2016*; *Mei et al., 2021*). These results demonstrated that the ER family has broad prospects for application in regulating plant development and stress tolerance.

In the current research, the characteristics of *SiER* family members (*SiER* s) in the foxtail millet were analyzed. *SiER1_X4*, and *SiER4_X1* genes were isolated, and their biomass and thermotolerance of transgenic *Arabidopsis* were evaluated. The findings provide the functional genes for potential use in improvement of production and stress resistance in gramineous crops.

## MATERIALS AND METHODS

### Phylogenetic analysis of the SiERs family in Setaria italica

Two *SiER* gene tags from foxtail millet (Seita.4G086700.1 and Seita.1G338900.1) were obtained with the sorghum *SbER10_X1* gene (XM_002437978.2) as a reference sequence after BLAST in the Phytozome v12.1 database. Four families of *SiER* members were obtained by searching the NCBI database with the two *SiER* tags to predict the complete CDS and chromosome-position information. The exon distribution (GSDS 2.0), *cis*-regulatory elements of promoters (Plant CARE), subcellular localization characteristics (Plant-mPLoc) and motif structure (MEME) of *SiERs* family were predicted. Moreover, the conserved functional domains (PROSITE and SMART databases), amino acid size, molecular weight, and isoelectric point (ProtParam) of SiERs proteins were analyzed. Table S1 lists all databases and their URLs available at the journal's website.

Based on the functional domains of SiERs, the amino acid sequences of the published ER family in monocot and dicot plants with similarity above 80% were downloaded from NCBI database (Annex S2, *SiER1_X4* gene was listed in Annex S4), to produce a SiERs phylogenetic tree by MEGA5.0 software with a threshold of 1,000 replications for bootstrap, according to the neighbor-joining method (*Tamura et al., 2013*).

## Genes isolation and subcellular localization of SiER1_X4 and SiER4_X1

Due to the abundant transcription of *SiERs* in the pedicel tissue of the Dungu variety at the heading stage, total RNA from pedicel was extracted with RNAprep Pure Kit (DP432; Beijing, China), and cDNA was synthesized with a PrimeScript First-Strand cDNA Synthesis Kit (6110A; Shiga, Japan). Taking the pedicel cDNA as material, specific primers (*SiER1_X4-F2/SiER1_X4-R2* and *SiER4_X1-F3/SiER4_X1-R3* in Annex S3) were designed to separate *SiER1_X4* and *SiER4_X1* fragment, respectively. The PCR reaction (50 µL) was as follows: 25 µL of 2 × PCR buffer, 10 µL of dNTP (2 mM), 1.5 µL of Primer-F (10 µM), 1.5 µL of Primer-R (10 µM), 1 µL of KOD FX (1.0 U/mL, KFX-101; Toyobo, China), 5 µL of cDNA as template, 6 µL of ddH$_2$O. The PCR procedure was as follows: 94 °C for 2 min, 40 cycles (98 °C for 10 s, 65 °C for renaturation in both *SiER1_X4* and *SiER4_X1* gene, lasting for 30 s, 68 °C for 4 min for extension), and 68 °C for 10 min.

The code fragment of *SiER1_X4* and *SiER4_X1* (without the stop codon) was separated through *SiER1_X4-gfpF1/SiER1_X4-gfpR1* and *SiER4_X1-gfpF1 /SiER4_X1-gfpR1* primers (Annex S3). The same PCR procedure and reaction system as above were used, except for the 62 °C and 61 °C for renaturation in *SiER1_X4-gfp* and *SiER4_X1-gfp* gene, respectively. The fusion-protein was generated as below: PCR products of *SiER1_X4* and *SiER4_X1* were differentially integrated into the N terminal of green fluorescent protein vector (pJIT16318-GFP), which included CaMV35S promoter. pJIT16318-SiER1_X4 and pJIT16318-SiER4_X1 were transferred into wheat mesophyll protoplasts (isolation from 10-day-old wheat seedlings) via the PEG4000-mediated method (*Cui et al., 2019*). The transformed cells were incubated at 22 °C in darkness for 18–20 h, and then observed and photographed under a confocal laser scanning microscope (LSM700; CarlZeiss, Germany).

## Thermotolerance identification of transgenic Arabidopsis

*SiER1_X4* and *SiER4_X1* segments (without the stop codon for fusion-protein development) were separated by primers of *SiER1_X4-1302F1/SiER1_X4-1302R1* and *SiER4_X1-1302F1/ SiER4_X1-1302R1*, respectively (Annex S3). The same PCR procedure and reaction system were used as above, except for 64 °C and 62 °C for renaturation in *SiER1_X4-1302* and *SiER4_X1-1302* gene, respectively. The PCR products of *SiER1_X4* and *SiER4_X1* were inserted into pCAMBIA1302 vector (CaMV35S promoter) to obtain the fusion vectors of pCAMBIA1302-SbER1_X4 and pCAMBIA1302-SbER4_X1, respectively. Using a *Agrobacterium tumefaciens*-mediated transformation system (*Bradley et al., 1997*), the targeted fusion vectors were transformed into *Arabidopsis* (Columbia ecotype). The offspring seeds were screened with antibiotics to obtain homozygous transgenic *SiER1_X4* and *SiER4_X1* lines. The test steps were described by *Chen et al. (2020)*.

The stable transgenic lines overexpressing the target genes were cultivated on MS medium for 3 days (without antibiotics), and then moved into a light incubator for 7

days. Seedlings of the similar size were transplanted into pots (6.8 × 6.8 cm) with nine plants in each pot and ten pots per transgenic line. After 10 days of continuous growth in a greenhouse (26 °C growth with an 8 h/16 h dark/light, photon flux density of 525 $\mu$mol m$^{-2}$ s$^{-1}$), five pots per transgenic line were treated in a light incubator at 42 °C for 48 h and 60 h, and the five remaining pots were cultivated at 26 °C for later biomass investigation (control).

After the high-temperature treatment for 60 h, leaves of transgenic and wild-type (WT) lines were collected, and some samples were used to determine SOD and POD activity, as described by *Zheng et al. (2020)*, the remainder was quickly frozen in liquid nitrogen, and stored at −80 °C for qRT-PCR. The remaining treated lines were transferred to the greenhouse to control conditions (26 °C) for 11 days, to observe the recovery growth of *Arabidopsis* plants, the number of plants with green leaves was counted to assess the survival rate of transgenic and WT lines after high-temperature treatment. Four individual plants from each line were served as biological replicates.

### Plant material and hormone-induction treatment

Five foxtail millet germplasm varieties (Dabaigu, Dungu, Jingu21, Yugu1, and Kuanjiu) were pre-germinated for 4 days. Seedlings with the similar germination were transplanted to pots (35 × 35 cm) with forty plants in each pot, and the flower pots were placed in a light incubator for growth (humidity 60%; temperature 23 °C/20 °C day/night; 16 h/8 h light/dark; light intensity 525 $\mu$mol m$^{-2}$ s$^{-1}$). After 6 days, mixture of the stems and leaves from a single plant for each variety was collected. After culturing the remaining plants for 15 days, seedlings were removed along with the roots, rinsed off the soil, and placed briefly on filter paper to dry, and then cultured in hormone solution or deionized water (control). The concentrations of the hormone solution were as follows: abscisic acid (ABA) 100 $\mu$M, brassinolides (BRs) 0.75 $\mu$M, gibberellin (GA$_3$) 30 mM and indole acetic acid (IAA) 10 $\mu$M (*Zheng & Hu, 2016*). Samples (mixture of stems and leaves) were separately collected for qRT-PCR. The treatment periods were 0, 1, 2, 4, 6, 12, 24, 48, and 60 h.

In May 2021, the Dungu variety was planted in the experimental field, and embryo and coleoptile were collected at the germination stage. Roots, stems, flag leaves, flag leaf sheaths, pedicels, and inflorescence samples were collected at the flowering stage. Seeds were collected at the maturity stage. All samples were quickly frozen in liquid nitrogen after collection and stored at −80 °C for later detection of *SiERs* expression patterns in diverse organs. Three individual plants were selected as biological replicates for each sample collection.

### qRT-PCR analysis

Nine cDNA sequences of the *SiERs* family were aligned to design specific primers for *SiER1_X4* and *SiER4_X1* qRT-PCR expression. The high-temperature related gene, *AtHSFA1a*, and superoxide suppressor gene, *AtBl1*, were used to determine the molecular-response mechanism of *SiER1_X4* and *SiER4_X1* in transgenic *Arabidopsis* plants after high-temperature stress (*Yoshida et al., 2011*; *Ishikawa, Uchimiya & Kawai-Yamada, 2013*). The primers of *SiER1_X4* (*SiER1_X4–qRTF2/SiER1_X4–qRTR2*), *SiER4_X1* (*SiER4_X1–qRTF1/ SiER4_X1–qRTR1*), *AtHSFA1a* (*AtHSFA1a-qRTF2/AtHSFA1a-qRTR2*), and *AtBl1*

(*AtBI1-qRTF1/AtBI1-qRTR1*), as well as the reference genes (*SiActin -qRT F1/SiActin -qRT R1* and *AtActin-qRTF5/ AtActin-qRTR5*), are listed in Annex S3. The target-gene-expression level was detected by qRT-PCR analysis with the ABI Prism 7500 system (Applied Biosystems, Waltham, MA, USA). Three technical replicates and three biological replicates were conducted for all experiments, and the $2^{-\Delta\Delta Ct}$ method was used for quantification (*Liu et al., 2013*).

## Data processing and statistical analysis

qRT-PCR data was analyzed in accordance with the procedure of *Zheng & Hu (2016)*. Error analysis was conducted with SPSS Statistics Software version 18.0 (SPSS18.0, IBM, USA) based on the biological replicates of three individual plants. The related indicators of agronomic traits were also statistically analyzed using SPSS18.0 software. The data of all graphs was represented as the mean ± standard error. The graphics were analyzed and produced with OriginPro 2018C SR1 and Excel 2010 software.

# RESULTS

## Characteristics and phylogenetic relationship of the SiERs of foxtail millet

Four genes were found in the *SiERs* family of foxtail millet. Among them, *SiERL4* (gene ID: LOC101753243) and *SiER4* (gene ID: LOC10175555 8097) were distributed on chromosome 4, and *SiER1* was located on chromosome 1 with two genes (gene ID: LOC101780996 and gene ID: LOC117840131) (Table 1). Further analysis (Fig. 1) showed that 1 copy and 26 exons were found in *SiERL4* sequences (XM_004964364.4), and 2 copies and 27 exons in *SiER4* sequences. In exon 25, 6 amino acids fewer were encoded in *SiER4_X2* (XM_004964885.3) than in *SiER4_X1* (XM_004964884.4). Three copies were found in the LOC101780996 gene of *SiER1*, exons 1 and 2 were lacking in *SiER1_X3* (XM_014804622.2), 22 exons were found in the other two copies, 5 amino acids were lacking in exon 20 of *SiER1_X2* (XM_014804623.2), and valine was lacking in exon 21 of *SiER1_X1* (XM_014804625.2). Three copies were found in the LOC117840131 gene of *SiER1*, each of which contained 27 exons, compared with *SiER1_X5* (XM_034720593.1), and mutations were found in exon 9 and 25 of *SiER1_X4* (Annex S4), and one amino acid was lost in exon 26 of *SiER1_X6* (XM_034720600.1). The amino acid structure prediction indicated that the proteins of the SiER4 family were larger, and the LOC101780996 of SiER1 was smaller. The nine copies of four genes in the SiERs family were all predicted to be transmembrane proteins, a typical feature of ER family proteins. In total 15 LRR tandem regions were detected in the SiERL4 protein, 13 LRR regions in SiER4, 9 LRR regions in LOC101780996 (SiER1), and 14 LRR regions in LOC117840131 (SiER1) (Annex S5).

In the published ER family, cluster analysis showed four categories (Fig. 2): Category I and Category II contained the monocotyledonous plants, with the six copies of SiER1 family and rice ER protein constituting the first category, in which SiER1_X1 and SiER1_X6, and SiER1_X4 and SiER1_X5 were closely related. Category II was composed of two copies of SiER4 family, as well as ERs of sorghum, maize, goatgrass, wheat, barley, and brachypodium. SiER4 family was closely related to sorghum and maize. Category III was composed of ER

Zheng et al. (2022), *PeerJ*, DOI 10.7717/peerj.14452

**Table 1** The characteristics of putative *SiER* genes in *Setaria italica* L.

| Name | Nucleotide | | | | | Protein | | | | Location |
|------|-----------|--|--|--|--|--------|--|--|--|----------|
| | Gene ID (NCBI) DNA | Gene length (bp) | Locus (NCBI) mRNA | Number of Exons | Protein accession (NCBI) | pI | Subcellular Location | Length of Protein (AA) | MW (KDa) | |
| *SiER1_X1* | | | XM_014804625.2 | 22 | XP_014660111.1 | 5.77 | Cell membrane | 786 | 86 | Chr. **I** |
| *SiER1_X2* | LOC101780996 | 5847 | XM_014804623.2 | 22 | XP_014660109.1 | 5.83 | Cell membrane | 782 | 85 | Chr. **I** |
| *SiER1_X3* | | | XM_014804622.2 | 20 | XP_014660108.1 | 5.90 | Cell membrane | 713 | 78 | Chr. **I** |
| *SiER1_X4* | | | Annex 4 (OP492075) | 27 | Annex 4 listing | 5.45 | Cell membrane | 991 | 108 | Chr.**I** |
| *SiER1_X5* | LOC117840131 | 7319 | XM_034720593.1 | 27 | XP_034576484.1 | 5.50 | Cell membrane | 991 | 108 | Chr. **I** |
| *SiER1_X6* | | | XM_034720600.1 | 27 | XP_034576491.1 | 5.50 | Cell membrane | 990 | 108 | Chr. **I** |
| *SiER4_X1* | LOC101758097 | 7761 | XM_004964884.4 | 27 | XP_004964941.1 | 5.87 | Cell membrane. | 997 | 109 | Chr.IV |
| *SiER4_X2* | | | XM_004964885.3 | 27 | XP_004964942.1 | 5.90 | Cell membrane. | 991 | 108 | Chr. IV |
| *SiERL4* | LOC101753243 | 6793 | XM_004964364.4 | 26 | XP_004964421.1 | 5.55 | Cell membrane. | 979 | 106 | Chr. IV |

**Notes.**

pI is isoelectric point; MW is the molecular weight of amino acids. OP492075 is a GenBank accession number for *SiER1_X4*.
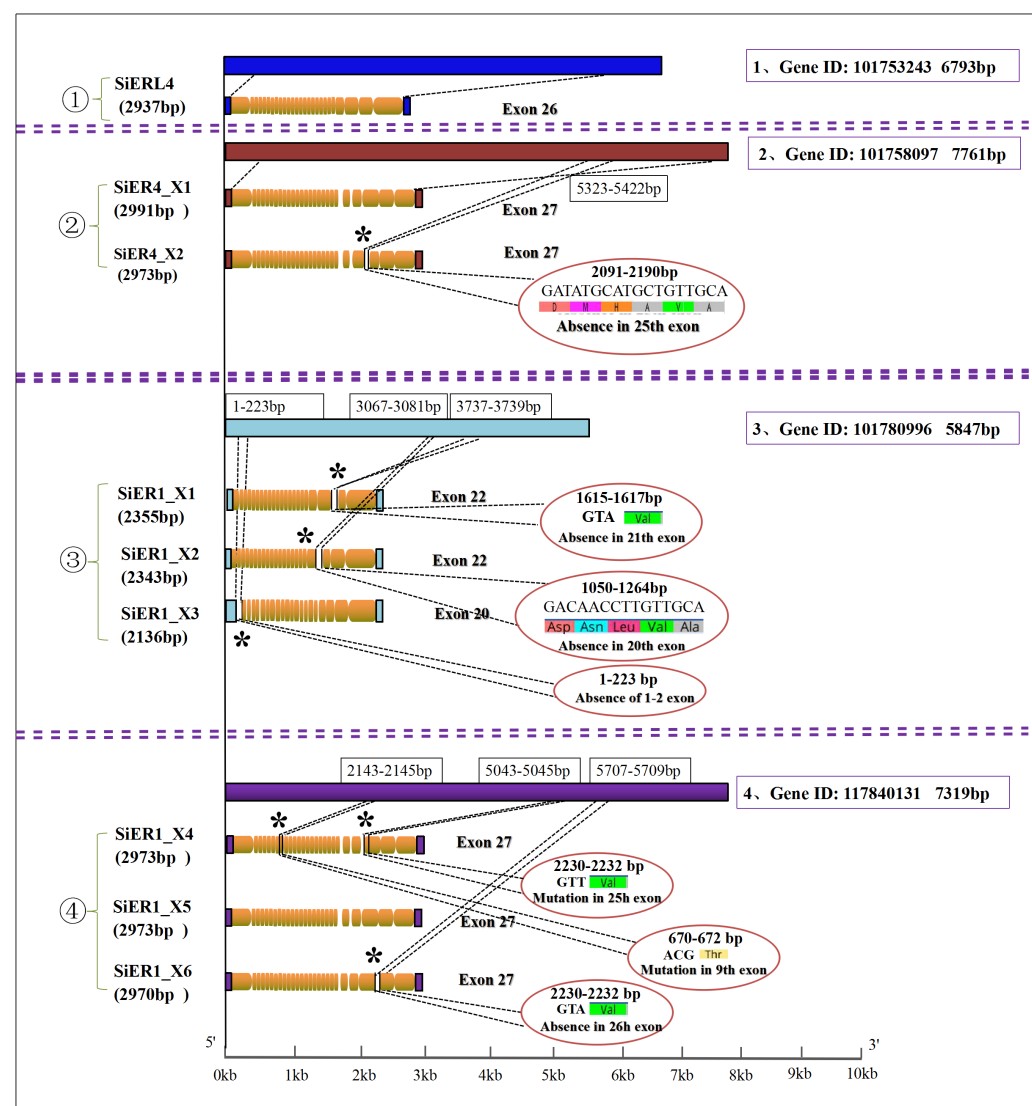

**Figure 1** **Nucleotide sequence characteristics of *SiER* family genes.** An asterisk (*) represents the mutation locus of amino acids.

family of dicotyledon as soybean and grape. Category IV was constituted by SiERL4 and *Arabidopsis* AtER and AtERL. These findings showed that in the evolution of ER families of different species, ERL was a seperate branching direction, the phylogenetic relationship of SbER1 family was close to modern aquatic plants, whereas that of SbER4 family was closer to field xerophytic plants.

## *SiERs* gene structure and its cis-regulatory elements

The *cis*-regulatory elements of *SiERs* family promoters were primarily involved in regulating three types of plant functional responses as follows (Table 2): (a) cell development process, including seed development, endosperm formation, meristem and mesophyll cell differentiation, cell-development cycle changes; (b) hormone-response mechanisms,

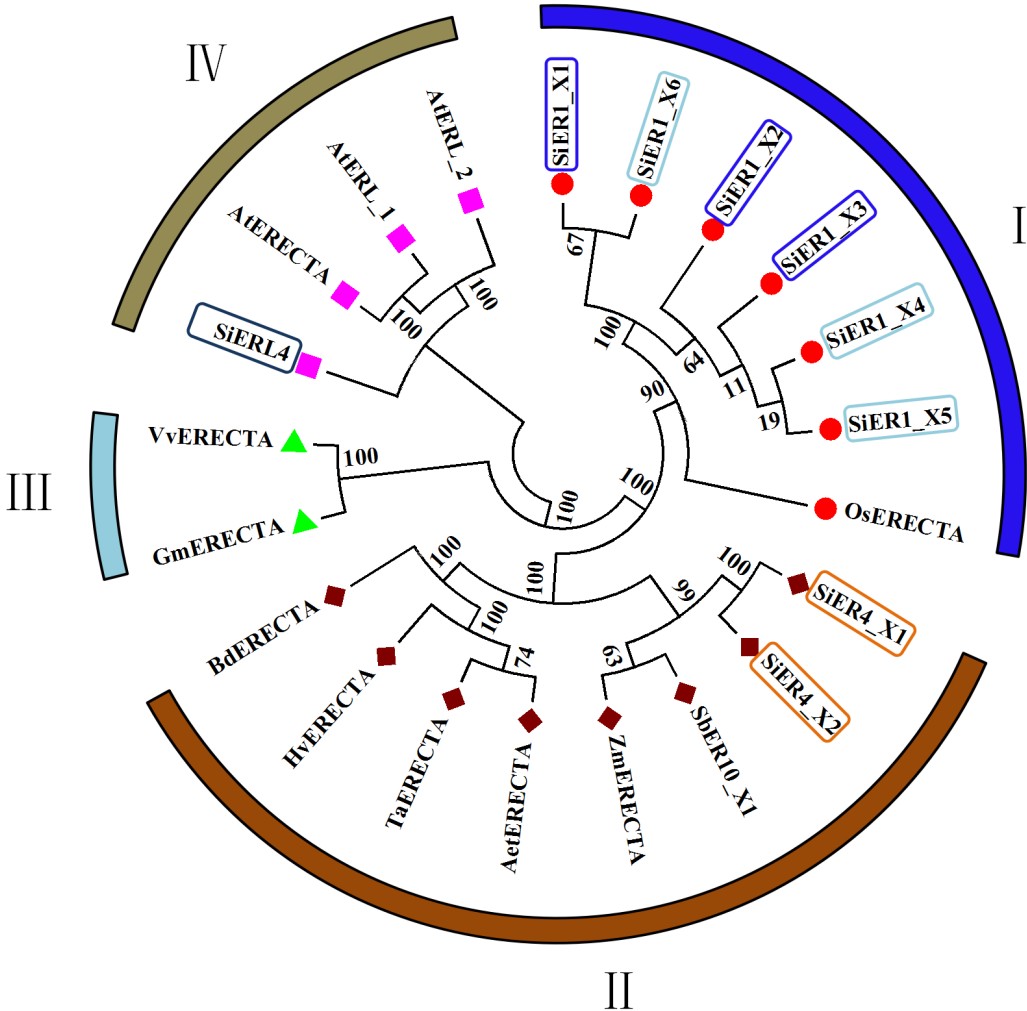

**Figure 2 Phylogenetic tree of ER family proteins in monocots and dicots.** Each category is represented by the same symbol with the same color. Numbers beside the branches represent bootstrap values based on 1,000 replications. Plant species and NCBI accession numbers of proteins in phylogenetic tree are listed in Annex S2.

including regulation pathways mediated by salicylic acid, methyl jasmonate, abscisic acid, gibberellin and auxin; (c) biological metabolic reactions, including light response, drought and low temperature induction, adversity defense, anaerobic induction, circadian rhythm regulation. These finding suggests that the SiERs could participate in the regulation of plant growth and development, and may increase plant resistance to external stress.

The SiERs is a typical receptor-like kinase (Annex S5), including the N-terminal signal-peptide region, the leucine tandem region (LRRs), the transmembrane region, and the C-terminal serine/threonine kinase domain. ER families of different species greatly differed in amino acid residues in the N-terminal signal-peptide region and transmembrane region (Annex S6). The 15 motif-conserved structures in the SiERs family can be divided into two categories (Fig. 3): The first category included SiER1_X1, SiER1_X2, and SiER1_X3,

**Table 2** Functional characteristic of *cis*-acting elements of *SiERs* promoters in *Setaria italica* L.

| Code | Functional elements of SiER promoters | | | | Functional characteristic | Note |
|---|---|---|---|---|---|---|
| | SiER1 (LOC101780996) | SiER1 (LOC117840131) | SiER4 (LOC101758097) | SiERL4 (LOC101753243) | | |
| 1 | RY-element | | RY-element | RY-element | Seed-specific regulation | Cell development process |
| 2 | | | | GCN4_motif | Endosperm expression | |
| 3 | CAT-box | | CAT-box | CAT-box | Meristem expression | |
| 4 | | | | HD-Zip 1 | The palisade mesophyll cells differentiation | |
| 5 | MSA-like | MSA-like | MSA-like | | Involved in cell cycle regulation | |
| 6 | TCA-element | | TCA-element | TCA-element, | Salicylic acid responsiveness | Hormone-response mechanisms |
| 7 | GARE-motif, TATC-box | | GARE-motif, TATC-box | P-box | Gibberellin responsive | |
| 8 | TGACG-motif, CGTCA-motif | TGACG-motif, CGTCA-motif | CGTCA-motif, TGACG-motif, | CGTCA-motif, TGACG-motif | Methyl jasmonate responsiveness | |
| 9 | ABRE | ABRE | ABRE | ABRE | Abscisic acid responsiveness | |
| 10 | TGA-element | | TGA-element | | Auxin responsive | |
| 11 | | | | TC-rich repeats | Defense and stress responsiveness | Biological metabolic reactions |
| 12 | Box 4, Sp1, GTGGC-motif, G-Box, TCCC-motif, GATA-motif, TCT-motif, ATCT-motif, GT1-motif | G-Box, Gap-box, GTGGC-motif, GT1-motif, TCCC-motif, | Box4, GT1-motif, G-Box, GTGGC-motif, Sp1, ATCT-motif, GATA-motif, TCCC-motif, TCT-motif, | TCCC-motif, Sp1, Box4, TCT-motif, L-box, G-Box, 3-AF1 binding site | Light responsive | |
| 13 | MBS | MBS | MBS | MBS | Drought inducibility | |
| 14 | LTR | LTR | LTR | LTR | Low temperature responsiveness | |
| 15 | ARE | ARE | ARE, GC-motif | ARE | The anaerobic induction | |
| 16 | GC-motif | GC-motif | | GC-motif | Anoxic specific inducibility | |
| 17 | Circadian | | Circadian | | Element involved in circadian control | |

**Notes.**
Functional characteristics of *cis*-acting elements of *SbER* promoters were predicted in the PlantCARE database (http://bioinformatics.psb.ugent.be/webtools/plantcare/html/).

whereas the remaining six copies were classified into the second category. In the first category, motif 14 and 13, encoding the N-terminal signal-peptide region and the 1-3 LRR

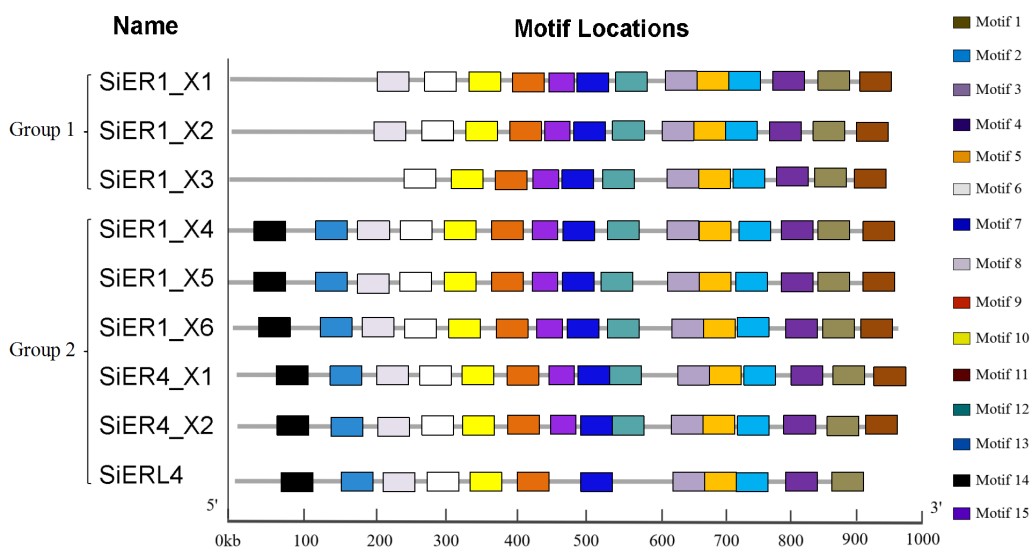

**Figure 3** Motif analysis of SiERs amino acid sequence.

tandem domains, respectively, were lacking. Motif 8, encoding No. 4 and 5 of the LRR region, was additionally lacking in SiER1_X3. In the second category, except for SiERL4 that lacked motif 15 and 12 (encoding 13-14 LRR structures and transmembrane region, respectively), the other SiER proteins were all equipped with 15 completely conserved motif structures. This finding showed that no significant difference existed in the motif distribution of SiER family members, except for some amino acid change during the SiERs evolution, indicating that the function of SiERs could be conserved in the foxtail millet.

The gene-structure characteristics of different *SiERs* copies revealed the following (Fig. 4): *SiERs* exons differed in length: exons 25, 26, and 27 near the 3′-UTR region were larger, which encoded the threonine/serine kinase region of ER proteins. Exons near the 5′-UTR region had different cascade numbers, which mainly encoded the leucine tandem region of ER proteins. From these characteristics, it was speculated that SiERs proteins had similar regulatory functions, which received upstream signal and transmitted them into the cell, to induce downstream genes effects by phosphorylation. In the LOC101780996 genes, *SiER1_X3* lacked the first two exons, and the distribution of other exons was similar. The first intron of *SiER4* family (LOC101758097) was larger, resulting in the largest sequence of *SiER4* family. *SiERL4* (LOC101753243) had 26 exons and was divided into a separate branch. It was reported that the ER family often constituted 27 exons, and *ERL* belonged to the *ERECTA-LIKE1* family (*Masle, Gilmore & Farquhar, 2005*; *Pillitteri & Torii, 2012*). In this study, both of *SiER1_X4* and *SiER4_X1* had 27 exons, showed typical gene-structure of the *SiERs* family, and were selected to to determine their functional characteristics.

## Expression patterns of SiERs in different foxtail millet varieties and diverse organs

Among the five common foxtail millet varieties in China, *SiER1_X4* and *SiER4_X1* showed the highest expression levels in Dungu, whereas *SiER1_X4* showed the lowest expression

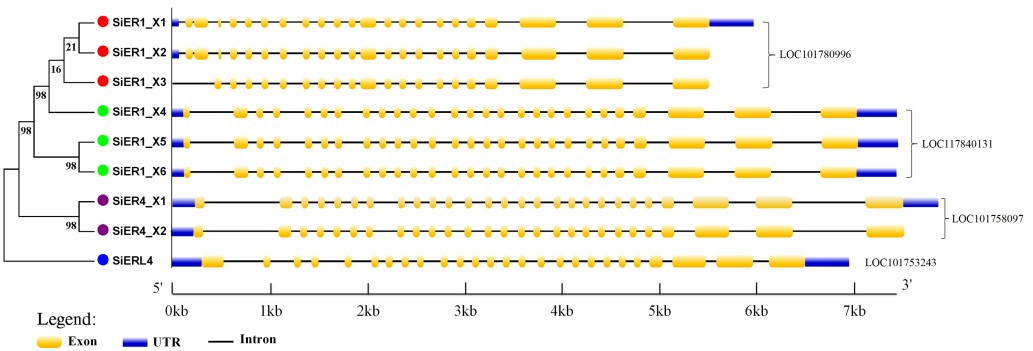

**Figure 4 Intron–exon structure of *SiERs* family in monocots and dicots.** Each gene is represented by the same symbol with the same color. Numbers beside the branches represent the bootstrap values based on 1,000 replications.

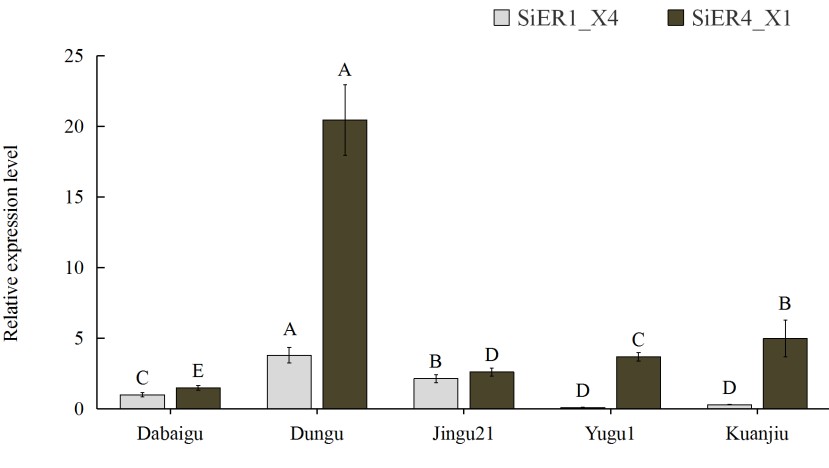

**Figure 5 Expression profiles of *SiER1_X4* and *SiER4_X1* gene in five varieties of foxtail millet ($n = 9$).** The capital letters represent the greatly significant difference of the same gene expression among different foxtail millet varieties ($P < 0.01$). The primers of *SiER1_X4* gene (*SiER1_X4–qRTF2/SiER1_X4–qRTR2*), *SiER4_X1 gene* (*SiER4_X1–qRTF1/SiER4_X1–qRTR1*) and reference gene (*SiActin-qRTF1/SiActin-qRTR1*) are listed in Annex S3.

level in Yugu 1, as well as the lowest expression level of *SiER4_X1* in Dabaigu (Fig. 5). Compared with *SiER1_X4*, *SiER4_X1* showed a higher expression level in the five foxtail millet varieties. This finding showed that *SiERs* had different transcription levels in different foxtail millet varieties and *SiER4_X1* may have a stronger regulatory function on the development of foxtail millet. Dungu was selected as an important material for subsequent gene-expression analysis.

In the different organs of Dungu, *SiER1_X4* and *SiER4_X1* genes were highly expressed in above-ground organs but rarely expressed in underground roots (Fig. 6). Taking root organ as a reference, the expression level of the two genes in the pedicel were both the
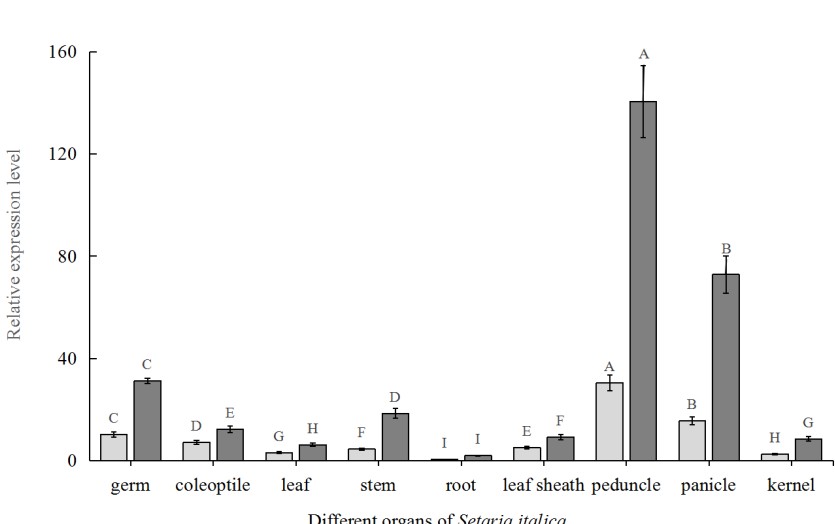

**Figure 6 Expression profiles of *SiER1_X4* and *SiER4_X1* genes during foxtail millet growth stages (*n* = 9).** Foxtail millet variety Dungu cDNA was used to detect expression patterns of the two genes. The capital letters represent the greatly significant difference of the same gene expression among different foxtail millet growth stages (*P* < 0.01). The primers of *SiER1_X4* gene (*SiER1_X4–qRTF2/SiER1_X4–qRTR2*), *SiER4_X1* gene (*SiER4_X1–qRTF1/SiER4_X1–qRTR1*) and reference gene (*SiActin-qRTF1/SiActin-qRTR1*) are listed in Annex S3.

highest, reaching 70 and 61 times of that in the roots, respectively. The expression level in panicle ranked the second (only 36 and 31 times, respectively). The expression levels in leaves and kernels were similar, both of which were at a low level. Thus, the functional roles of *SiERs* probably differed in regulating the development of different organs of foxtail millet, and the transcription levels of *SiER4_X1* gene in different organs were significantly higher than those of *SiER1_X4*.

## Expression patterns of SiER1_X4 and SiER4_X1 under hormone induction and subcellular localization analysis

Upon treatments with the hormones abscisic acid, brassinolides, gibberellin, and indole acetic acid, *SiER1_X4* and *SiER4_X1* established stable expression levels in the respective control samples, whereas a significantly increased expression level was observed in the treated samples (*P* < 0.01). With prolonged hormone-treatment time, the expression levels of the two genes showed a response pattern of initial increase and then decrease (Fig. 7). After treatment with ABA, the expression levels of the two genes rapidly increased. At 2 h, the expression reached the highest level, those of *SiER1_X4* and *SiER4_X1* were 7.1 and 8.6 times of the respective controls, respectively. After treatment with BRs for 2 h, the expression levels of *SiER1_X4* and *SiER4_X1* gene gradually increased, the expression was the highest at 6 h. Upon treatment with GA$_3$, the expression levels of the two genes rapidly increased after 2 h, and the expression was the highest at 4 h, which were 15.9 and 7.0 times of the control, respectively, after which the expression level rapidly decreased. After auxin (IAA) treatment, the expression of *SiER1_X4* slowly increased, whereas the expression

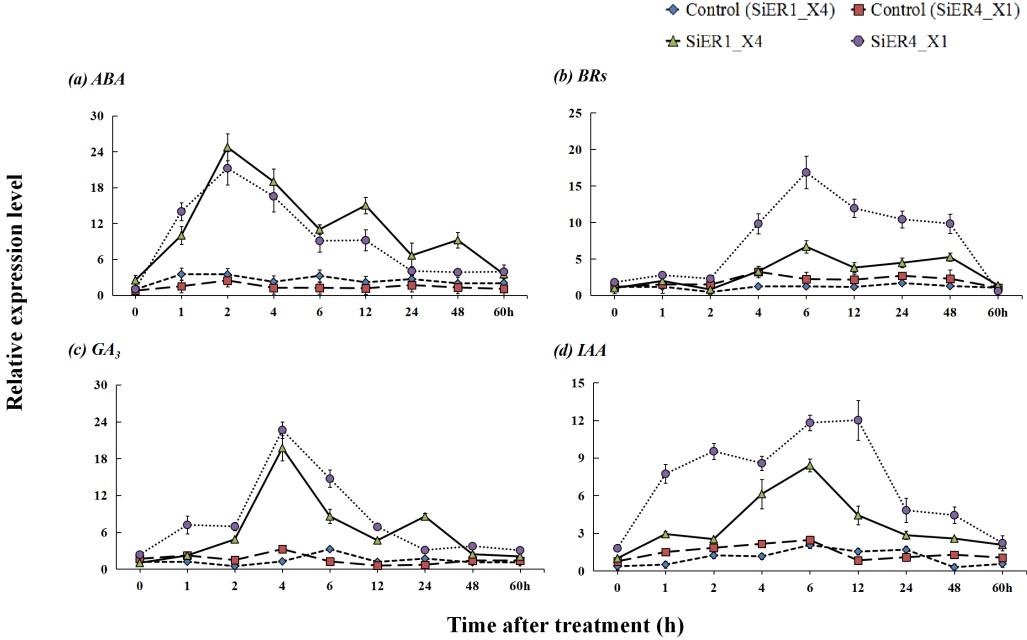

**Figure 7** **Expression patterns of *SiER1_X4* and *SiER4_X1* genes after hormone induction.** Foxtail millet variety Dungu cDNA was used to detect expression patterns of the two genes ($n = 9$). (A) Abscisic acid (ABA) treatment (100 µM); (B) brassinolide (BR) treatment (0.75 µM); (C) gibberellin (GA$_3$) treatment (30 mM); and (D) auxin (IAA) treatment (10 µM).

of *SiER4_X1* rapidly increased. At 6 h and 12 h respectively, the expression of the two genes reached their highest levels, respectively. Thus, compared with IAA treatment, the transcription level of the *SiER4_X1* gene was higher under the other three treatments. These findings showed that SiERs actively respond to hormone induction and might participate in the regulation of millet development and stress-resistance related physiological processes.

The ORF fragments of *SiER1_X4* and *SiER4_X1* were 2973 bp and 2991 bp, respectively (Annex S7). The subcellular localization analysis showed that the fluorescence signals of the two fusion proteins were located on the cell membrane and chloroplast of wheat mesophyll protoplasts, whereas the control pJIT16318-GFP was distributed on the cell membrane, cytoplasm and nucleus (Fig. 8). This result indicated that SiER1_X4 and SiER4_X1 primarily acted on cell membranes and chloroplasts, which was consistent with the above-mentioned prediction of SiERs as transmembrane proteins.

## Overexpression of SiERs in Arabidopsis thaliana increased the biomass

*SiER1_X4* and *SiER4_X1* were transformed into *Arabidopsis*, and the T$_4$ generation plants were investigated (Fig. 9). In the transgenic lines *OxSiER1_X4#3* and *OxSiER4_X1#13*, the expression levels of *SiER1_X4* and *SiER4_X1* were 66 and nine times those of control lines (WT), respectively. Compared with WT, the plant height of the two transgenic lines significantly increased ($P < 0.01$), and the main stem diameter and the biomass per plant were significantly higher than those of WT lines ($P < 0.05$), indicating that

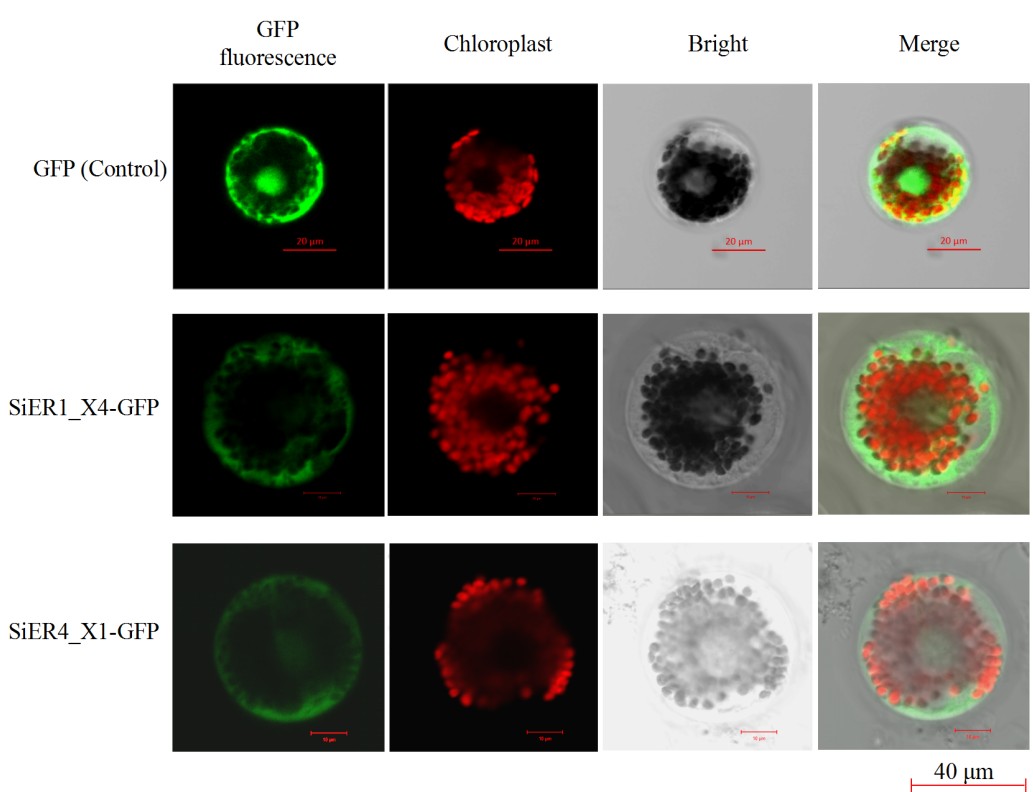

**Figure 8  Subcellular localization of SiER1_X4 and SiER4_X1 fusion proteins in wheat mesophyll protoplasts.** SbER1_X4-GFP, SbER4_X1-GFP, and pJIT16318-GFP (control) were transiently expressed in wheat mesophyll protoplasts, respectively. Images were captured using a confocal microscope (scale bar = 40 μm).

overexpression of *SiER1_X4* and *SiER4_X1* gene could enhance the biomass of *Arabidopsis*. It had significant implications for improving the biomass of forage crops, such as sorghum and foxtail millet. Among them, the total number of siliques per plant of the *SiER1_X4* transgenic lines was significantly more than those of WT lines ($P < 0.05$), whereas silique number was only slightly increased for *SiER4_X1* lines. Meanwhile, plant height, total number of siliques per plant, and biomass per plant of *SiER1_X4* transgenic *Arabidopsis* were higher than those of *SiER4_X1* transgenic lines.

## Thermotolerance of *Arabidopsis thaliana* overexpressing SiERs genes

After treating *Arabidopsis* overexpressing *SiER1_X4* and *SiER4_X1* genes at elevated temperature (42 °C), the plant leaves withered and several plants showed local necrosis. After recovering for 11 days at 26 °C, only a few plants of WT lines showed vital signs, and the others all died; whereas the survival rate of transgenic *Arabidopsis* was extremely and significantly higher than that of WT ($P < 0.01$), expecially the *SiER4_X1* transgenic plants, which showed a stronger ability to restore growth (Figs. 10A and 10B). Further determination of the antioxidant-enzyme activity of *Arabidopsis* showed that the SOD activity of *SiER1_X4* and *SiER4_X1* lines before and after high-temperature treatment was

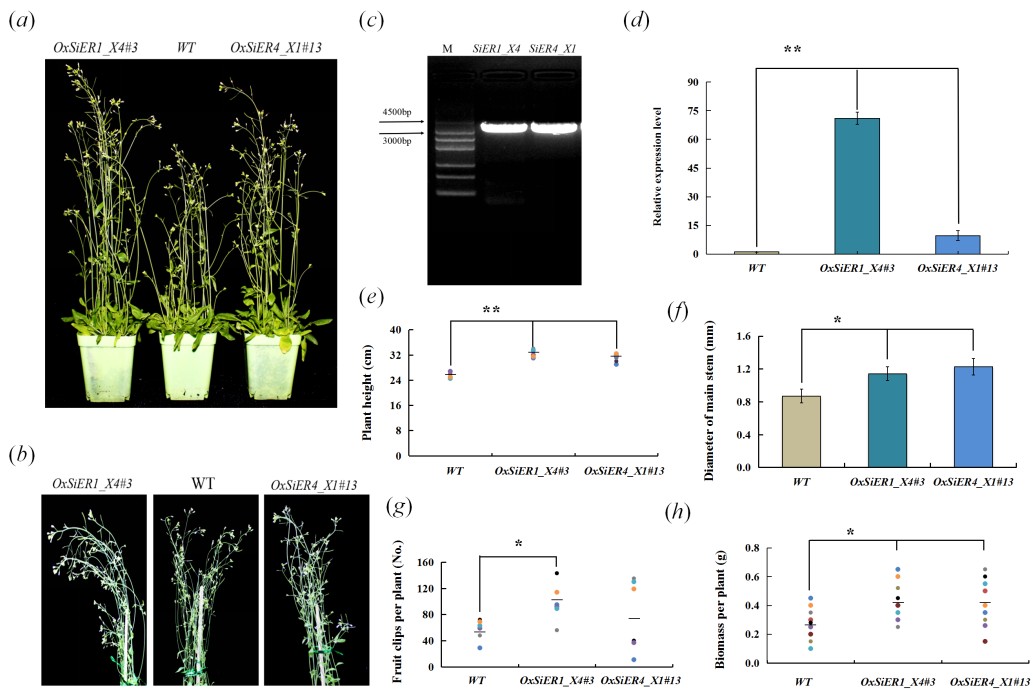

**Figure 9** **Biomass-related traits of transgenic *Arabidopsis*.** WT is the wild *Arabidopsis* lines, *OxSiER4_X1#13* and *OxSiER1_X4#3* are *Arabidopsis* lines transfected from *SiER4_X1* and *SiER1_X4* genes, respectively. (A) *Arabidopsis* plants grown for 30 days; (B) plant stalk of *Arabidopsis* grown for 30 days; (C) the fragment isolated from *SiER1_X4* and *SiER4_X1* genes (Annex S7); (D) detection of overexpression level of transgenic *Arabidopsis* ($n = 9$); (E) plant height of transgenic *Arabidopsis* ($n = 6$); (F) main stem diameter of transgenic *Arabidopsis* ($n = 7$); (G) total number of siliques per plant of transgenic *Arabidopsis* ($n = 6$); and (H) biomass per plant of transgenic *Arabidopsis* ($n = 9$). Asterisks represent a significant difference (* $P < 0.05$; ** $P < 0.01$).

significantly higher than that of WT plants, as well as the POD activity of both transgenic lines ($P < 0.01$) (Fig. 10C). Under high-temperature stress, the SOD and POD activities of *SiER4_X1* lines were slightly higher than those of *SiER1_X4* lines.

Analysis of expression of the high-temperature regulation gene, *AtHSF1*, and the superoxide suppressor gene, *AtBl1*, showed that the expression level of *AtHSF1* in the transgenic lines was extremely and significantly higher than those of WT lines ($P < 0.01$) (Fig. 10D). Particularly, after high-temperature induction, the *AtHSF1* expression level of transgenic lines significantly increased. Before high-temperature treatment, the expression level of *AtBl1* did not significantly differ between the transgenic lines and WT. After high-temperature treatment, the expression level of *AtBl1* increased and reached a significant difference in the *SiER4_X1* lines ($P < 0.05$). Before and after high-temperature treatment, the *AtHSF1* expression level of WT lines did not change significantly, whereas the expression level of *AtBl1* significantly increased ($P < 0.05$). These findings suggested that overexpression of *SiER1_X4* and *SiER4_X1* genes may improve the high-temperature tolerance of *Arabidopsis*, which may be due to the influence of heat-related gene expression

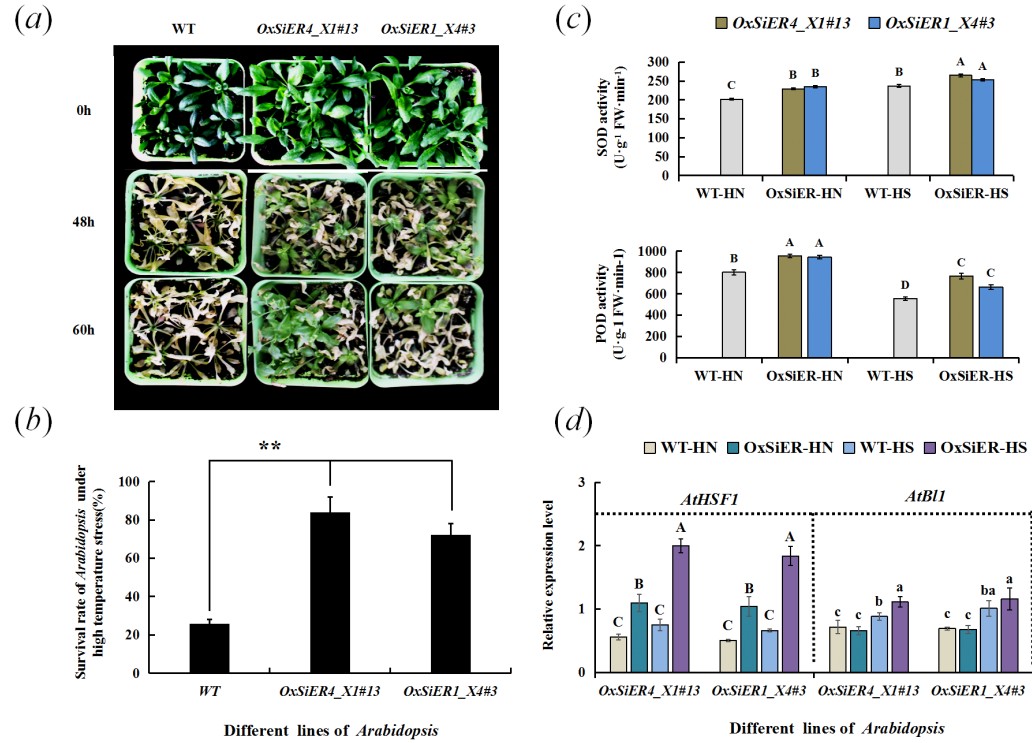

**Figure 10 Detection of thermotolerance of transgenic *Arabidopsis*.** WT is the wild type of *Arabidopsis* lines, *OxSiER4_X1#13* and *OxSiER1_X4#3* are *Arabidopsis* lines transfected from *SiER4_X1* and *SiER1_X4* genes, respectively. HN and HS represent well-culture and high-temperature stress plants, respectively. (A) Restored culturing for 11 d after high-temperature stress of transgenic *Arabidopsis*; (B) survival rate of transgenic *Arabidopsis* after high-temperature stress ($n = 5$); (C) SOD and POD activity of transgenic *Arabidopsis* ($n = 4$); and (D) expression identification of *AtHSF1* and *AtBl1* gene in transgenic *Arabidopsis* ($n = 9$). Capital and lowercase letters represent a significant difference at 0.01 and 0.05 level, respectively.

in the regulatory pathway and induction of the variable activity of related antioxidant enzymes. Moreover, *SiER4_X1* showed a better regulatory function than *SiER1_X4*.

## DISCUSSION

The characteristics of gene families have become an important means to analyze their function. The accuracy and reliability of analysis on the evolutionary features depend on genome-sequencing information. This study found that the foxtail millet genome contained four *SiER* family members, two genes were distributed on the first chromosome, with a total of six copies, and two genes were distributed on the fourth chromosome, with three copies. In rice, wheat, sorghum, cotton, tobacco crops, the ER family also had two members, and each member had different spliceosomes, resulting in an uneven distribution of the number of introns and exons in the genome (*Liu et al., 2019*). In foxtail millet, the spliceosomes in different copies of *SiERs* had obvious different forms, indicating that the relationship of the *SiERs* family was more complicated in the evolutionary process. In eukaryotes, the gain or loss of introns is one of the evolutionary mechanisms of creation of a gene family

(*Roy & Penny, 2007*), and furthermore, the difference in the number of introns affected the target-gene expression level. The introns of *AtER* genes were absent in *Arabidopsis*, leading to the reduced target protein by 500–900 times (*Karve et al., 2011*). With decreased LRR in the extracellular region of soybean GmER (decreased exons), shading treatment increased the hypocotyl length, leaf area, and petiole length of *Arabidopsis* (*Du et al., 2018*). We speculate that different spliceosomes of *SiERs* result in differences in regulatory functions.

Before the emergence of monocotyledonous and dicotyledonous plants, the ER family evolved into two large subfamilies, namely, ER and ERL. Later, with the occurrence of gene-replication events, multiple copies of ER and ERL families gradually formed (*Liu et al., 2019*). In the present study, the ER family can be clearly divided into four categories: aquatic monocot, terrestrial monocot, dicot, and *Arabidopsis* ER and ERL families. Among them, six copies on the first chromosome were closely related to aquatic monocots (rice), and two copies on the fourth chromosome were closely related to terrestrial monocots. Further analysis of amino acid sequences of SiERs in other species showed that different ER families greatly differed in amino acid residues in the N-terminal signal-peptide recognition and transmembrane regions. ER family proteins are transmembrane proteins that can sense external stimuli, activate the expression of intracellular signal factors, and regulate the physiological response of cells (*Shpak et al., 2004*). The most important function of ER was phosphorylation. The amino acid position difference in the transmembrane region influenced the phosphorylation event, and the N-terminal extension region was one of the components of overall kinase folding that was critical to the kinase activity (*Kosentka et al., 2017*).

The ER family was reported to involve in light-induced under growth (*Van Zanten et al., 2010*), improve drought resistance of maize (*Li et al., 2019*), and participate in the regulation of non-host resistance of rice blast disease, and coordinately regulate the resistance of *Arabidopsis* to the quantitative traits of *Verticillium* wilt together with ABA and methyl jasmonate (*Häffner et al., 2014*). Moreover, it inhibited cell division and promote cell elongation (*Qu, Zhao & Tian, 2017*). As determined in the current research, *SiER* promoters contained core elements related to abscisic acid, low temperature, drought, methyl jasmonate, anaerobic induction, and light response, suggesting that SiERs may played an important roles in plant resistance and photosynthesis. However, no study has reported regarding the mechanism of low-temperature and anaerobic-induced responses. *Van Zanten et al. (2009)* also reported that ER affected the photoelectron-transfer capacity and carboxylation rate of ribulose diphosphate carboxylase (Rubisco), thereby increasing the photosynthetic capacity of *Arabidopsis*. SiERs had two common *cis*-acting elements, G-Box and TCCC-motif, which were involved in the light-response process. Moreover, SiER1_X4 and SiER4_X1 were both located on chloroplasts, implying that SiERs are involved in the photosynthetic function. These results indicated great application potential for improving foxtail millet photosynthesis and plant biomass.

Overexpression of *SiER* s could promote *Arabidopsis* biomass accumulation, which was primarily due to the increase of stem thickness and plant height of transgenic plants, whereas the amount of pod numbers was uncertain. This finding was similar to previous results (*Xing et al., 2011*; *Masle, Gilmore & Farquhar, 2005*). Under high-temperature treatment,

*Arabidopsis* overexpressing *SiERs* had strong survival ability, and the SOD activity of transgenic lines significantly increased. Increased SOD activity could eliminate the damage to cells inflicted by reactive oxygen species produced by plants under high-temperature stress (*De-Pinto, Locato & De-Gara, 2012*). *SiERs* may be involved in the regulation of SOD synthesis or activity at high temperature and alleviate the damage to cells inflicted by $O^{2-}$ and $H_2O_2$ during adversity. Moreover, the expression levels of the high-temperature regulation gene *AtHSF1* and superoxide suppressor gene *AtBl1* confirmed the above statement, and the specific mechanism requires further study. In *Arabidopsis*, transforming CERK1n-ERc complex factors showed that under high-temperature stress, the $H_2O_2$ and related electrolyte content in transgenic *Arabidopsis* were less, and the ability to withstand high temperatures was significantly increased (*Chen et al., 2020*). This finding was similar to our current results, thereby providing an important basis for the next step to reveal the molecular mechanism of high-temperature tolerance of crops.

## CONCLUSIONS

This study analyzed the characteristics of *SiER* family members (*SiER* s) in foxtail millet. The foxtail-millet genome contained four *SiERs* member. Among them, *SiER1_X4* and *SiER4_X1* actively responded to the induced reaction of ABA, BRs, $GA_3$, and IAA, with a higher expression level in above-ground organs of foxtail millet. Compared to wild type, the transgenic *Arabidopsis* lines overexpressing the two genes enhanced the plant height and biomass accumulation, and showed the higher SOD and POD activities under high temperature, reflecting an increased thermotolerance in *Arabidopsis* plants. These results provided potential targets for conventional breeding or biotechnological methods to improve forage crop production under harsh environments.

## ACKNOWLEDGEMENTS

We acknowledge professor Malcolm Hawkesford in Rothamsted Research and Yanlong Liu of Anhui Science and Technology University who conceived of the experiments and improved the English grammar and spelling. We are grateful to the technical help of many post-graduate students, in particular Jie Yu, Hanjing Dai, Hong Zhang, Luqiao Qian. We are grateful to Ph.D Yusheng Wang in Shanxi Agricultural University for help with the foxtail millet varieties.

### Funding

This research was financially supported by the General Project of Anhui Science and Technology University (2021zryb15), the National Natural Science Foundation of China (31971993), and the Key Project of Natural Science Research of the Anhui Provincial Education Department (KJ2020A0066). The funders had no role in study design, data collection and analysis, decision to publish, or preparation of the manuscript.

## Grant Disclosures

The following grant information was disclosed by the authors:

General Project of Anhui Science and Technology University: 2021zryb15.

National Natural Science Foundation of China: 31971993.

Key Project of Natural Science Research of the Anhui Provincial Education Department: KJ2020A0066.

## Competing Interests

Haizhou Chen is employed by Anhui Youxin Agricultural Science and Technology Co. LTD.

## Author Contributions

- Jiacheng Zheng conceived and designed the experiments, performed the experiments, analyzed the data, prepared figures and/or tables, authored or reviewed drafts of the article, and approved the final draft.
- Xiaoyi Huang conceived and designed the experiments, performed the experiments, analyzed the data, prepared figures and/or tables, authored or reviewed drafts of the article, and approved the final draft.
- Jieqin Li conceived and designed the experiments, performed the experiments, authored or reviewed drafts of the article, and approved the final draft.
- Qingyuan He conceived and designed the experiments, performed the experiments, analyzed the data, authored or reviewed drafts of the article, and approved the final draft.
- Wan Zhao conceived and designed the experiments, authored or reviewed drafts of the article, and approved the final draft.
- Chaowu Zeng conceived and designed the experiments, performed the experiments, authored or reviewed drafts of the article, and approved the final draft.
- Haizhou Chen conceived and designed the experiments, performed the experiments, authored or reviewed drafts of the article, and approved the final draft.
- Qiuwen Zhan conceived and designed the experiments, analyzed the data, authored or reviewed drafts of the article, and approved the final draft.
- Zhaoshi Xu conceived and designed the experiments, authored or reviewed drafts of the article, and approved the final draft.

## Data Availability

The raw data is available in the Supplemental Files.

## Supplemental Information

Supplemental information for this article can be found online at http://dx.doi.org/10.7717/peerj.14452#supplemental-information.

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
