# Peer review of "Enhanced biomass and thermotolerance of Arabidopsis by SiERECTA isolated from Setaria italica L"

_PeerJ, doi:10.7717/peerj.14452_

## Round 0.1 · original submission · Major Revisions

Please address the queries and comments raised by the reviewers, especially in the annotated file.

Reviewer 1 ·

Basic reporting

Authors have selected ERECTA genes from foxtail millet and did the bioinformatic analysis to determine the structural and functional characteristics of the gene. They studied the expression levels of these genes in different organs of the plant. They cloned two different SiER genes from foxtail millet and overexpressed in A. thaliana and studied the effect of these genes in improvement of high temperature tolerance and improved plant biomass.

The article is well written and authors have provided all the necessary data files for the review. However, two tables were missing and authors are advised to submit the files for the review. The manuscript needs thorough grammatical and spell check. Authors need to check for the punctuations used all throughout the manuscript.

Experimental design

At certain sections, authors have missed certain details of the experiment conducted. All the suggestions and comments are included in the edited version of the manuscript.

Validity of the findings

In one of their study, authors have used 35S promoter to observe the site-specific expression of their GFP fused protein. As this is a constitutive promoter, it would not provide the actual expression of the gene and authors need to express the GFP fusion gene under a native promoter to understand its site-specific expression.

Annotated reviews are not available for download in order to protect the identity of reviewers who chose to remain anonymous.

Reviewer 2 ·

Basic reporting

In the current study, the authors analyzed the SiER family members (SiERs) in the foxtail millet genome and characterized the SiERs family regulatory elements and evolutionary relationship using genome-sequencing information. Accuracy and reliability of the analysis of the evolutionary features of gene families have become important resources to analyze their function. The study revealed that the foxtail-millet genome contained four SiERs members, two genes were distributed on the first chromosome and the other two are on the fourth chromosome. Further, the identified SiER1_X4, and SiER4_X1 genes were overexpressed in Arabidopsis thaliana and showed that the two genes enhanced the plant height and biomass accumulation. The transgenic lines also showed higher SOD and POD activities under high temperatures, reflecting an increased thermotolerance in Arabidopsis.

Global warming increases average temperatures and causes severe heat waves, challenging plant growth, and agriculture worldwide. Foxtail millet is an important C4 panicoid crop in arid and semi-arid regions of Asia and Africa due to its strong tolerance to drought stresses. In addition, the availability of genome sequence databases has encouraged the foxtail millet research community to perform high throughput investigations in the aspects of functional genomics. However, these resources are not widely used in the functional analysis of important genes. The authors wisely compared transcriptomic changes among monocotyledons (foxtail millet, Arabidopsis) under abiotic stress to gain insights into mechanisms related to biomass and high-temperature tolerance.
The author’s findings are excellent resources for potential breeding targets or biotechnological methods to improve the potential production and stress resistance in gramineous crops.

Minor comment:
Grammar needs to be improved throughout the manuscript. The authors need to address it before they publish.

Experimental design

Methods described with sufficient detail & information to replicate.

Validity of the findings

Conclusions are well stated, linked to original research question & limited to supporting results.

---

## Round 0.2 · Minor Revisions

The authors have addressed the Reviewers' comments and queries. The manuscript can now be accepted after some grammatical and spelling corrections, as raised by one of the Reviewers. Please go through the manuscript thoroughly and improve the grammar and spelling.

Reviewer 1 ·

Basic reporting

Authors have revised the manuscript as per the recommendations and the revisions are satisfactory. However, authors are advised to check for any grammatical and spelling corrections all throughout the manuscript. Few minor corrections are included in the edited version of the manuscript.

Experimental design

No comments

Validity of the findings

No comments

Annotated reviews are not available for download in order to protect the identity of reviewers who chose to remain anonymous.

Reviewer 2 ·

Basic reporting

The authors addressed the grammatical errors in the revised manuscript. The manuscript can be accepted for publication.

Experimental design

Methods described with sufficient detail & information to replicate.

Validity of the findings

Conclusions are well stated, linked to original research question .

---

## Round 0.3 · accepted · Accept

Thank you for making the suggested edits and improving the manuscript for publcation. I did notice a few typos that I am recommending be changed in the published version:

Last sentence of the manuscript, instead of "inteventio" I think yiou mean "intervention"

Line 174, should be "hat the proteins of the SiER4 family were . . . "

Line 237, "above-ground"

Please let us know if these corrections are in error.